# Credal Self-Supervised Learning

**Julian Lienen**
Department of Computer Science
Paderborn University
Paderborn 33098, Germany
`julian.lienen@upb.de`

**Eyke Hüllermeier**
Institute of Informatics
University of Munich (LMU)
Munich 80538, Germany
`eyke@ifi.lmu.de`

## Abstract

Self-training is an effective approach to semi-supervised learning. The key idea is to let the learner itself iteratively generate "pseudo-supervision" for unlabeled instances based on its current hypothesis. In combination with consistency regularization, pseudo-labeling has shown promising performance in various domains, for example in computer vision. To account for the hypothetical nature of the pseudo-labels, these are commonly provided in the form of probability distributions. Still, one may argue that even a probability distribution represents an excessive level of informedness, as it suggests that the learner precisely knows the ground-truth conditional probabilities. In our approach, we therefore allow the learner to label instances in the form of credal sets, that is, sets of (candidate) probability distributions. Thanks to this increased expressiveness, the learner is able to represent uncertainty and a lack of knowledge in a more flexible and more faithful manner. To learn from weakly labeled data of that kind, we leverage methods that have recently been proposed in the realm of so-called superset learning. In an exhaustive empirical evaluation, we compare our methodology to state-of-the-art self-supervision approaches, showing competitive to superior performance especially in low-label scenarios incorporating a high degree of uncertainty.

## 1 Introduction

Recent progress and practical success in machine learning, especially in deep learning, is largely due to an increased availability of data. However, even if data collection is cheap in many domains, labeling the data so as to make it amenable to supervised learning algorithms might be costly and often comes with a significant effort. As a consequence, many data sets are only partly labeled, i.e., only a few instances are labeled while the majority is not. This is the key motivation for semi-supervised learning (SSL) methods [7], which seek to exploit both labeled and unlabeled data simultaneously.

As one simple yet effective methodology, so-called *self-training* [26], often also referred to as pseudo-labeling, has proven effective in leveraging unlabeled data to improve over training solely on labeled data. The key idea is to let the learner itself generate "pseudo-supervision" for unlabeled instances based on its own current hypothesis. In the case of probabilistic classifiers, such pseudo-targets are usually provided in the form of (perhaps degenerate) probability distributions. Obviously, since pseudo-labels are mere guesses and might be wrong, this comes with the danger of biasing the learning process, a problem commonly known as confirmation bias [46]. Therefore, self-training is nowadays typically combined with additional regularization means, such as consistency regularization [2, 40], or additional uncertainty-awareness [38].

Labeling an instance $x$ with a probability distribution on the target space $\mathcal{Y}$ is certainly better than committing to a precise target value, for example a single class label in classification, as the latter would suggest a level of conviction that is not warranted. Still, one may argue that even a probability distribution represents an excessive level of informedness. In fact, it actually suggests that the learner

35th Conference on Neural Information Processing Systems (NeurIPS 2021).

precisely knows the ground-truth conditional probability $p(y \mid \boldsymbol{x})$. In our approach, we therefore allow the learner to label instances in the form of *credal sets*, that is, sets of (candidate) probability distributions [27]. Thanks to this increased expressiveness, the learner is able to represent uncertainty and a lack of knowledge about the true label (distribution) in a more flexible and more faithful manner. For example, by assigning the biggest credal set consisting of all probability distributions, it is able to represent complete ignorance — a state of knowledge that is arguably less well represented by a uniform probability distribution, which could also be interpreted as full certainty about this distribution being the ground truth. Needless to say, this ability is crucial to avoid a confirmation bias and account for the heteroscedastic nature of uncertainty, which varies both spatially (i.e., among different regions in the instance space) and temporally: typically, the learner is less confident in early stages of the training process and becomes more confident toward the end.

Existing methods are well aware of such problems but handle them in a manner that is arguably ad-hoc. The simple yet effective SSL framework FixMatch [41], for instance, applies a thresholding technique to filter out presumably unreliable pseudo-labels, which often results in unnecessarily delayed optimization, as many instances are considered only lately in the training. Other approaches, such as MixUp [53], also apply mixing strategies to learn in a more cautious manner from pseudo-labels [1, 3, 4]. Moreover, as many of these approaches, including FixMatch, rely on the principle of entropy minimization [19] to separate classes well, the self-supervision is generated in the form of rather peaked or even degenerate distributions to learn from, which amplifies the problems of confirmation bias and over-confidence [33].

An important implication of credal pseudo-labeling is the need for extending the underlying learning algorithm, which must be able to learn from weak supervision of that kind. To this end, we leverage the principle of generalized risk minimization, which has recently been proposed in the realm of so-called superset learning [22]. This approach supports the idea of *data disambiguation*: The learner is free to (implicitly) choose any distribution inside a credal set that appears to be most plausible in light of the other data (and its own learning bias). Thus, an implicit trade-off between cautious learning and entropy minimization can be realized: Whenever it seems reasonable to produce an extreme distribution, the learner is free but not urged to do so. Effectively, this not only reduces the risk of a potential confirmation bias due to misleading or over-confident pseudo-labels, it also allows for incorporating all unlabeled instances in the learning process from the beginning without any confidence thresholding, leading to a fast and effective semi-supervised learning method.

To prove the effectiveness of this novel type of pseudo-labeling, we proceed from FixMatch as an effective state-of-the-art SSL framework and replace conventional probabilistic pseudo-labeling by a credal target set modeling. In an exhaustive empirical evaluation, we study the effects of this change compared to both hard and soft probabilistic target modeling, as well as measuring the resulting network calibration of induced models to reflect biases. Our experiments not only show competitive to superior generalization performance, but also better calibrated models while cutting the time to train the models drastically.

## 2   Related Work

In semi-supervised learning, the goal is to leverage the potential of unlabeled in addition to labeled data to improve learning and generalization. As it constitutes a broad research field with a plethora of approaches, we will focus here on classification methods as these are most closely related to our approach. We refer to [7] and [48] for more comprehensive overviews.

As one of the earliest ideas to incorporate unlabeled data in conventional supervised learning, *self-training* has shown remarkably effective in various domains, including natural language processing [12] and computer vision [11, 17, 39]. The technique is quite versatile and can be applied for different learning methods, ranging from support vector machines [30] to decision trees [45] and neural networks [35]. It can be considered as *the* basic training pattern in distillation, such as self-distillation from a model to be trained itself [23] or within student-teacher settings [36, 51]. Recently, this technique also lifted completely unsupervised learning in computer vision to a new level [6, 20].

As a common companion of self-training for classification, especially in computer vision, *consistency regularization* is employed to ensure similar model predictions when facing multiple perturbed versions of the same input [2, 37, 40], resulting in noise-robustness as similarly achieved by other (stochastic) ensembling methods such as Dropout [42]. Strong augmentation techniques used in this

regard, e.g., CTAugment [3] or RandAugment [9], allow one to learn from instances outside of the (hitherto labeled) data distribution and, thus, lead to more accurate models [10]. For semi-supervised learning in image classification, the combination of consistency regularization with pseudo-labeling is widely adopted and has proven to be a simple yet effective strategy [1, 4, 25, 37, 41, 50, 55].

As pseudo-labeling comprises the risk of biasing the model by wrong predictions, especially when confidence is low, uncertainty awareness has been explicitly considered in the generic self-supervision framework UPS to construct more reliable targets [38]. Within their approach, the model uncertainty is estimated by common Bayesian sampling techniques, such as MC-Dropout [15] or DropBlock [16], which is then used to sort out unlabeled instances for which the model provides uncertain predictions. Related to this, the selection of pseudo-labels based on the model certainty has also been used in specific domains, such as text classification [32] or semantic segmentation [54].

## 2.1 FixMatch

As already mentioned, FixMatch [41] combines recent advances in consistency regularization and pseudo-labeling into a simple yet effective state-of-the-art SSL approach. It will serve as a basis for our new SSL method, as it provides a generic framework for fair comparisons between conventional and our credal pseudo-labeling.

In each training iteration, FixMatch considers a batch of $B$ labeled instances $\mathcal{B}_l = \{(\boldsymbol{x}_i, p_i)\}_{i=1}^{B} \subset \mathcal{X} \times \mathbb{P}(\mathcal{Y})$ and $\mu B$ unlabeled instances $\mathcal{B}_u = \{\boldsymbol{x}_i\}_{i=1}^{\mu B} \subset \mathcal{X}$, where $\mathcal{X}$ denotes the input feature space, $\mathcal{Y}$ the set of possible classes, $\mathbb{P}(\mathcal{Y})$ the set of probability distributions over $\mathcal{Y}$, and $\mu \geq 1$ the multiplicity of unlabeled over labeled instances in each batch. Here, the probabilistic targets in $\mathcal{B}_l$ are given as degenerate "one-hot" distributions. When aiming to induce probabilistic classifiers of the form $\hat{p} : \mathcal{X} \longrightarrow \mathbb{P}(\mathcal{Y})$, FixMatch distinguishes between two forms of augmentation: While $\mathcal{A}_s : \mathcal{X} \longrightarrow \mathcal{X}$ describes "strong" augmentations that perturbs the image in a drastic manner by combining multiple operations, simple flip-and-shift transformations are captured by "weak" augmentations $\mathcal{A}_w : \mathcal{X} \longrightarrow \mathcal{X}$.

As an iteration-wise loss to determine gradients, a combination of the labeled loss $\mathcal{L}_l$ and unlabeled loss $\mathcal{L}_u$ is calculated. For the former, the labeled input instances from $\mathcal{B}_l$ are weakly augmented and used in a conventional cross-entropy loss $H : \mathbb{P}(\mathcal{Y})^2 \longrightarrow \mathbb{R}$. For the latter, the model prediction $q := \hat{p}(\mathcal{A}_w(\boldsymbol{x}))$ on a weakly-augmented version of each unlabeled instance $\boldsymbol{x}$ is used to construct a (hard) pseudo-label $\tilde{q} \in \mathbb{P}(\mathcal{Y})$ when meeting a predefined confidence threshold $\tau$. While $\tilde{q}$ is in FixMatch a degenerate probability distribution by default, one could also inject soft probabilities, which, however, turned out to be less effective (cf. [41]). The pseudo-label is then compared to a strongly-augmented version of the same input image. Hence, the unlabeled loss $\mathcal{L}_u$ is given by

$$\mathcal{L}_u := \frac{1}{\mu B} \sum_{\boldsymbol{x} \in \mathcal{B}_u} \mathbb{I}_{\max q \geq \tau} H(\tilde{q}, \hat{p}(\mathcal{A}_s(\boldsymbol{x}))) \ .$$

## 3 Credal Self-Supervised Learning

In this section, we introduce our credal self-supervised learning (CSSL) framework for the case of classification, assuming the target to be categorical with values in $\mathcal{Y} = \{y_1, \ldots, y_K\}$. Before presenting more technical details, we start with a motivation for the credal (set-valued) modeling of target values and a sketch of the basic idea of our approach.

### 3.1 Motivation and Basic Idea

In supervised learning, we generally assume the dependency between instances $\boldsymbol{x} \in \mathcal{X}$ and associated observations (outcomes) to be of stochastic nature. More specifically, one typically assumes a "ground-truth" in the form of a conditional probability distribution $p^*(\cdot \mid \boldsymbol{x}) \in \mathbb{P}(\mathcal{Y})$. Thus, for every $y \in \mathcal{Y}$, $p^*(y \mid \boldsymbol{x})$ is the probability to observe $y$ as a value for the target variable in the context $\boldsymbol{x}$. Ideally, the distribution $p^*(\cdot \mid \boldsymbol{x})$ would be provided as training information to the learner, along with every training instance $\boldsymbol{x}$. In practice, however, supervision comes in the form of concrete values of the target, i.e., a realization $y \in \mathcal{Y}$ of the random variable $Y \sim p^*(\cdot \mid \boldsymbol{x})$, and the corresponding degenerate distribution $p_y$ assigning probability mass 1 to $y$ (i.e., $p_y(y \mid \boldsymbol{x}) = 1$ and $p_y(y' \mid \boldsymbol{x}) = 0$ for $y' \neq y$) is taken as a surrogate for $p^*$.

Obviously, turning the true distribution $p^* = p^*(\cdot \mid \boldsymbol{x})$, subsequently also called a "soft label", into a more extreme distribution $p_y$ (i.e., a single value $y$), referred to as "hard label", may cause undesirable effects. In fact, it suggests a level of determinism that is not warranted and tempts the learner to over-confident predictions that are poorly calibrated, especially when training with losses such as cross-entropy [33]. So-called label smoothing [44] seeks to avoid such effects by replacing hard labels $p_y$ with (hypothetical) soft labels $\hat{p}$ that are "close" to $p_y$.

Coming back to semi-supervised learning, training of the learner and self-supervision can be characterized for existing methods such as FixMatch as follows:

- The learner is trained on hard labels, which are given for the labeled instances $\boldsymbol{x}_l$ and constructed by the learner itself for unlabeled instances $\boldsymbol{x}_u$.
- Construction of (pseudo-)labels is done in two steps: First, the true soft label $p^* = p^*(\cdot \mid \boldsymbol{x}_u)$ is predicted by a distribution $\hat{p} = \hat{p}(\cdot \mid \boldsymbol{x}_u)$, and the latter is turned into a hard label $\hat{p}_y$ afterward, provided $\hat{p}$ suggests a sufficient level of certainty (support for $y$).

This approach can be challenged for several reasons. First, training probabilistic predictors on hard labels comes with the disadvantages mentioned above and tends to bias the learner. Second, pseudo-labels $\hat{p}_y$ constructed by the learner tend to be poor approximations of the ground truth $p^*$. In fact, there will always be a discrepancy between $p^*$ and its prediction $\hat{p}$, and this discrepancy is further increased by replacing $\hat{p}$ with $\hat{p}_y$. Third, leaving some of the instances — those for which the prediction is not reliable enough — completely unlabeled causes a loss of information. Roughly speaking, while a part of the training data is overly precise[1], another part remains unnecessarily imprecise and hence unused. This may slow down the training process and cause other undesirable problems such as "path-dependency" (training is influenced by the order of the unlabeled instances).

To avoid these disadvantages, we propose to use soft instead of hard labels as training information for the learner. More specifically, to account for possible uncertainty about a true soft label $p^*$, we model information about the target in the form of a set $Q \subseteq \mathbb{P}(\mathcal{Y})$ of distributions, in the literature on imprecise probability also called a *credal set* [27]. Such a "credal label" is supposed to cover the ground truth, i.e., $p^* \in Q$, very much like a confidence interval in statistics is supposed to cover (with high probability) some ground-truth parameter to be estimated.

This approach is appealing, as it enables the learner to model its belief about the ground-truth $p^*$ in a cautious and faithful manner: Widening a credal label $Q$ may weaken the training information but maintains or even increases its validity. Moreover, credal labeling elegantly allows for covering the original training information as special cases: A hard label $y$ provided for a labeled instance $\boldsymbol{x}_l$ corresponds to a singleton set $Q_y = \{p_y\}$, and the lack of any information in the case of an unlabeled instance $\boldsymbol{x}_u$ is properly captured by taking $Q = \mathbb{P}(\mathcal{Y})$. Starting with this information, the idea is to modify it in two directions:

- Imprecisiation: To avoid possible disadvantages of hard labels $Q_y$, these can be made less precise through label relaxation [29], which is an extension of the aforementioned label smoothing. Technically, it means that $Q_y$ is replaced by a credal set $Q$ containing, in addition to $p_y$ itself, distributions $p$ close to $p_y$.
- Precisiation: The non-informative and maximally imprecise labels $Q = \mathbb{P}(\mathcal{Y})$ for unlabeled instances $\boldsymbol{x}_u$ are successively (iteration by iteration) "shrunken" and made more precise. This is done by replacing a set $Q$ with a smaller subset $Q' \subset Q$, provided the exclusion of certain candidate distributions $p \in Q$ is sufficiently supported by the learner.

### 3.2 Credal Labeling

Credal sets are commonly assumed to be convex, i.e., $p, q \in Q$ implies $\lambda p + (1 - \lambda)q \in Q$ for all distributions $p, q$ and $\lambda \in (0, 1)$. In our context, arbitrary (convex) credal sets $Q \subseteq \mathbb{P}(\mathcal{Y})$ could in principle be used for the purpose of labeling instances. Yet, to facilitate modeling, we restrict ourselves to credal sets induced by so-called *possibility distributions* [13].

A possibility or plausibility measure $\Pi$ is a set-function $2^{\mathcal{Y}} \longrightarrow [0, 1]$ that assigns a degree of plausibility $\Pi(Y)$ to every subset (event) $Y \subseteq \mathcal{Y}$. Such a measure is induced by a possibility

---

[1]To some extent, this can be alleviated through measures such as instance weighing, i.e., by attaching a weight to a pseudo-labeled instances [38].

distribution $\pi : \mathcal{Y} \longrightarrow [0,1]$ via $\Pi(Y) = \max_{y \in Y} \pi(y)$ for all $Y \subseteq \mathcal{Y}$. Interpreting degrees of plausibility as upper probabilities, the set of (candidate) probability distributions $p$ (resp. measures $P$) in accordance with $\pi$ resp. $\Pi$ is given by those for which $P(Y) \leq \Pi(Y)$ for all events $Y$:

$$Q_\pi = \left\{ p \in \mathbb{P}(\mathcal{Y}) \,|\, \forall Y \subseteq \mathcal{Y} : P(Y) = \sum_{y \in Y} p(y) \leq \max_{y \in Y} \pi(y) = \Pi(Y) \right\} . \tag{1}$$

Roughly speaking, for each $y \in \mathcal{Y}$, the possibility $\pi(y)$ determines an upper bound for $p^*(y)$, i.e., the highest probability that is deemed plausible for $y$. Note that, to guarantee $Q_\pi \neq \emptyset$, possibility distributions must be normalized in the sense that $\max_{y \in \mathcal{Y}} \pi(y) = 1$. In other words, there must be at least one outcome $y \in \mathcal{Y}$ that is deemed completely plausible.

In our context, this outcome is naturally taken as the label $y = \operatorname{argmax}_{y' \in \mathcal{Y}} \hat{p}(y')$ with the highest predicted probability. A simple way of modeling then consists of controlling the degree of imprecision (ignorance of the learner) by a single parameter $\alpha \in [0,1]$, considering credal sets of the form

$$Q_y^\alpha = \left\{ p \in \mathbb{P}(\mathcal{Y}) \,|\, p(y) \geq 1 - \alpha \right\} . \tag{2}$$

Thus, $Q_y^\alpha$ consists of all distributions $p$ that allocate a probability mass of at least $1 - \alpha$ to $y$ and hence at most $\alpha$ to the other labels $\mathcal{Y} \setminus \{y\}$. As important special cases we obtain $Q_y^0 = \{p_y\}$, i.e., the degenerate distribution that assigns probability 1 to the label $y$, and $Q_y^1 = \mathbb{P}(\mathcal{Y})$ modeling complete ignorance about the ground truth $p^*$.

Needless to say, more sophisticated credal sets could be constructed on the basis of a distribution $\hat{p}$, for example leveraging the concept of probability-possibility transformations [14]. Yet, to keep the modeling as simple as possible, we restrict ourselves to sets of the form (2) in this work.

## 3.3 Learning from Credal Labels

Our approach requires the learner to be able to learn from credal instead of probabilistic or hard labels. To this end, we refer to the generic approach to so-called superset learning as proposed in [22]. Essentially, this approach is based on minimizing a generalization $\mathcal{L}^*$ of the original (probabilistic) loss $\mathcal{L} : \mathbb{P}(\mathcal{Y})^2 \longrightarrow \mathbb{R}$. More specifically, the so-called *optimistic superset loss* [22], also known as *infimum loss* [5], compares credal sets with probabilistic predictions as follows:

$$\mathcal{L}^*(Q, \hat{p}) = \min_{p \in Q} \mathcal{L}(p, \hat{p}) \tag{3}$$

For the specific case where credal sets are of the form (2) and the loss $\mathcal{L}$ is the Kullback-Leibler divergence $D_{KL}$, (3) simplifies to

$$\mathcal{L}^*(Q_y^\alpha, \hat{p}) = \left\{ \begin{array}{cc} 0 & \text{if } \hat{p} \in Q_y^\alpha \\ D_{KL}(p^r || \hat{p}) & \text{otherwise} \end{array} \right. , \tag{4}$$

where

$$p^r(y') = \left\{ \begin{array}{cc} 1 - \alpha & \text{if } y' = y \\ \alpha \cdot \frac{\hat{p}(y')}{\sum_{y'' \neq y} \hat{p}(y'')} & \text{otherwise} \end{array} \right. \tag{5}$$

is the projection of the prediction $\hat{p}$ onto the boundary of $Q$. This loss has been proven to be convex, making its optimization practically feasible [29].

The loss (3) is an optimistic generalization of the original loss in the sense that it corresponds to the loss $\mathcal{L}(p, \hat{p})$ for the most favorable instantiation $p \in Q$. This optimism is motivated by the idea of *data disambiguation* [22] and can be justified theoretically [5]. Roughly speaking, minimizing the sum of generalized losses over all training examples $(\boldsymbol{x}_i, Q_i)$ comes down to (implicitly) choosing a precise probabilistic target inside every credal set, i.e., replacing $(\boldsymbol{x}_i, Q_i)$ by $(\boldsymbol{x}_i, p_i)$ with $p_i \in Q_i$, in such a way that the original loss (empirical risk) can be made as small as possible.

## 3.4 Credal Self-Supervised Learning Framework

Our idea of credal self-supervised learning (short CSSL) offers a rather generic framework for designing self-supervised learning methods. In this paper, we focus on image classification as a concrete and practically relevant application, and combine CSSL with the consistency regularization framework provided by FixMatch (cf. Fig. 1).

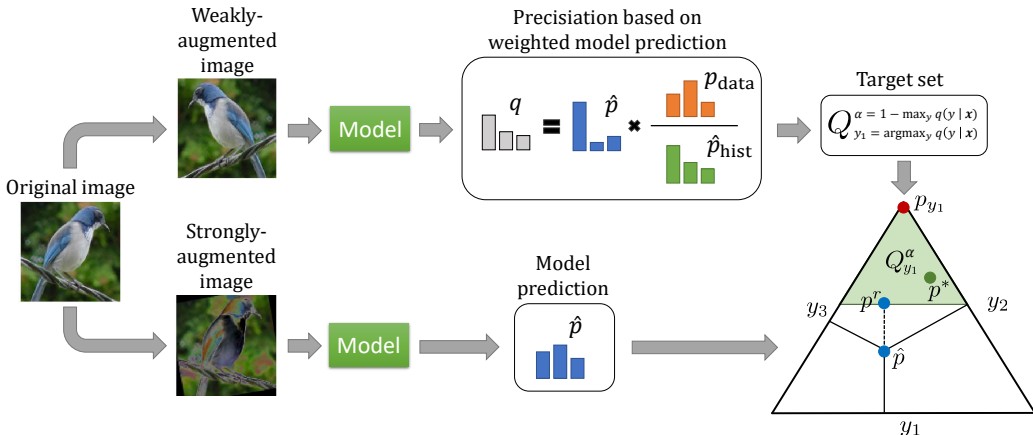

Figure 1: Schematic overview of the learning framework for unlabeled instances following [41]: Given classes $\mathcal{Y} = \{y_1, y_2, y_3\}$, a credal target set $Q$ is generated from the prediction on a weakly-augmented version of the input image. As illustrated in the barycentric coordinate system on the right bottom, $Q$ (red shaded region) covers the ground-truth distribution $p^*$ (green point). The degenerate distribution $p_{y_1}$ assigning probability 1 to $y_1$ corresponds to the red point on the top. To calculate the final loss, the model prediction $\hat{p}$ on a strongly-augmented version of the original image is compared to $Q$, whereby its projection onto $Q$ is depicted by $p^r$.

To describe the algorithm, we again assume batches of labeled $\mathcal{B}_l$ and unlabeled instances $\mathcal{B}_u$ (cf. Section 2.1). For the former, we measure the labeled loss $\mathcal{L}_l$ in terms of the cross-entropy $H$ between the observed target distributions of the (labeled) training instances and the current model predictions $\hat{p}$ on weakly-augmented features. At this point, one could apply the idea of imprecisiation through label relaxation as motivated in Section 3.1. However, as this work focuses on the self-supervision part, i.e., on $\mathcal{B}_u$ rather than $\mathcal{B}_l$, we stick to hard labels. This also facilitates the interpretation of experimental results later on, as it avoids the mixing of different effects.

For the unlabeled instances $\boldsymbol{x}_i \in \mathcal{B}_u$, we follow the consistency regularization idea of FixMatch and use the predictions on weakly augmented versions of the unlabeled instances as a reference for the target set construction. More precisely, we take the class $y_i = \mathrm{argmax}_{y \in \mathcal{Y}} \hat{p}_i(y)$ of the current model prediction $\hat{p}_i = \hat{p}(\mathcal{A}_w(\boldsymbol{x}_i)))$ as a reference to construct instance-wise targets $Q_{y_i}^{\alpha_i}$ as specified in (2).

A rather straightforward approach to determining the uncertainty level $\alpha_i$ is to set $\alpha_i = 1 - \hat{p}_i(y_i)$. Yet, motivated by the idea of distribution alignment as proposed in [3], we suggest to weight the predictions $\hat{p}_i$ by the proportion of the class prior $\tilde{p} \in \mathbb{P}(\mathcal{Y})$ and a moving average of the last model predictions $\bar{p} \in \mathbb{P}(\mathcal{Y})$, so that $\alpha_i = 1 - q_i(y) / \sum_{y' \in \mathcal{Y}} q_i(y')$ with the weighted (pseudo-)probability scores

$$q_i(y) = \hat{p}_i(y') \times \frac{\tilde{p}(y')}{\bar{p}(y')} \ . \tag{6}$$

According to this way of modeling pseudo-labels in terms of credal sets, the size (imprecision) of a set is in direct correspondence with the confidence of the learner. Therefore, this approach leads to some sort of (implicit) disambiguation: With increasing confidence, the target sets are becoming more precise, successively fostering entropy minimization without imposing overly constrained targets. Also, as mentioned before, this approach allows for using all instances for training from the very beginning, without losing any of them due to confidence thresholding. The weighting mechanism based on the class prior $\tilde{p}$ and the prediction history $\bar{p}$ (second factor on the right-hand side of (6)) accounts for the consistency and hence the confidence in a particular class prediction. If $\tilde{p}(y') \gg \bar{p}(y')$, the label $y'$ is under-represented in the past predictions, suggesting that its true likelihood might be higher than predicted by the learner, and vice versa in the case where $\tilde{p}(y') \ll \bar{p}(y')$. As this procedure is simple and computationally efficient, it facilitates the applicability compared to computationally demanding uncertainty methods such as MC-Dropout.

The proposed target sets are then used within the unlabeled loss $\mathcal{L}_u$ according to (4), which is used in addition to the labeled loss $\mathcal{L}_l$. Hence, the final loss is given by

$$\mathcal{L} = \underbrace{\frac{1}{|\mathcal{B}_l|} \sum_{(\boldsymbol{x}_i, p_i) \in \mathcal{B}_l} H(p_i, \hat{p}_i)}_{\mathcal{L}_l} + \lambda_u \underbrace{\frac{1}{|\mathcal{B}_u|} \sum_{\boldsymbol{x}_i \in \mathcal{B}_u} \mathcal{L}^*(Q_{y_i}^{\alpha_i}, \hat{p}_i)}_{\mathcal{L}_u} \ . \tag{7}$$

The pseudo-code of the complete algorithm can be found in the appendix.

We conclude this section with a few remarks on implementation details. Since we are building upon FixMatch, we keep the same augmentation policy as suggested by the authors. Thus, we employ CTAugment by default. We refer to the appendix for further ablation studies, including RandAugment as augmentation policy. Moreover, we consider the same optimization algorithm as used before, namely SGD with Nesterov momentum, for which we use cosine annealing as learning rate schedule [31]. Similar to FixMatch, we set the learning rate to $\eta \cos \frac{7\pi k}{16K}$, where $\eta$ is the initial learning rate, $k$ the current training step and $K$ the total number of steps ($2^{20}$ by default). As we are keeping the algorithmic framework the same and do not require any form of confidence thresholding, we can reduce the number of parameters compared to FixMatch, which further facilitates the use of this approach. We also use an exponential moving average of model parameters for our final model, which comes with appealing ensembling effects that typically improve model robustness and has been considered by various recent approaches for un- or semi-supervised learning [6, 41].

## 4 Experiments

To compare our idea of credal pseudo-labeling, we conduct an exhaustive empirical evaluation with common image classification benchmarks. More precisely, we follow the semi-supervised learning evaluation setup as described in [41] and perform experiments on CIFAR-10/-100 [24], SVHN [34], and STL-10 [8] with varying fractions of labeled instances sampled from the original data sets, also considering label-scarce settings with only a few labels per class. For CIFAR-10, SVHN, and STL-10, we train a Wide ResNet-28-2 [52] with 1.49 M parameters, while we consider Wide ResNet-28-8 (23.4 M parameters) models for the experiments on CIFAR-100. To guarantee a fair comparison to existing methods related to FixMatch, we keep the hyperparameters the same as in the original experiments. We refer to the appendix for a more comprehensive overview of the experimental details. We repeat each run 5 times with different seeds for a higher significance and average the results for the model weights of the last 20 epochs as done in [41].

As baselines, we report the results for Mean Teacher [46], MixMatch [4], UDA [50], ReMixMatch [3], and EnAET [49] as state-of-the art semi-supervised learning methods. Moreover, as we directly compete against the probabilistic hard-labeling employed in FixMatch, we compare our approach to this (with CTAugment) and related methods, namely AlphaMatch [18], CoMatch [28], and ReRankMatch [47]. Since these approaches extend the basic framework of FixMatch by additional means (e.g., loss augmentations), the direct comparison with FixMatch is maximally fair under these conditions, avoiding side-effects as much as possible. In addition, we show the results of the student-teacher method Meta Pseudo Labels (Meta PL) [36].

### 4.1 Generalization Performance

In the first experiments, we train the aforementioned models on the four benchmark data sets for different numbers of labeled instances to measure the generalization performance of the induced models. The results are provided in Table 1.

As can be seen, CSSL is especially competitive when the number of labels is small, showing that the implicit uncertainty awareness of set-based target modeling becomes effective. But CSSL is also able to provide compelling performance in the case of relatively many labeled instances, although not substantially improving over conventional hard pseudo-labeling. For instance, it is approximately on par with state-of-the-art performance on CIFAR-100 and SVHN. Focusing on the comparison to FixMatch, credal self-supervision improves the performance in almost all cases over hard pseudo-labeling with confidence thresholding, providing further evidence for the adequacy of our method.

Table 1: Averaged misclassification rates for 5 different seeds using varying numbers of labeled instances (**bold** font indicates the best performing method and those within two standard deviations per data set and label number). Approaches using different models, so that the comparison may not be entirely fair, are marked with ∗. We also show the results for STL-10 as reported in [18], using smaller models due to computational resource limitations, which we mark by †.

| | CIFAR-10 | | | CIFAR-100 | | | SVHN | | | STL-10† |
| | 40 lab. | 250 lab. | 4000 lab. | 400 lab. | 2500 lab. | 10000 lab. | 40 lab. | 250 lab. | 1000 lab. | 1000 lab. |
|---|---|---|---|---|---|---|---|---|---|---|
| Mean Teacher | - | 32.32 ±2.30 | 9.19 ±0.19 | - | 53.91 ±0.57 | 35.83 ±0.24 | - | 3.57 ±0.11 | 3.42 ±0.07 | - |
| MixMatch | 47.54 ±11.50 | 11.05 ±0.86 | 6.42 ±0.10 | 67.61 ±1.32 | 39.94 ±0.37 | 28.31 ±0.33 | 42.55 ±14.53 | 3.98 ±0.23 | 3.50 ±0.28 | 14.84 ±1.24 |
| UDA | 29.05 ±5.93 | 8.82 ±1.08 | 4.88 ±0.18 | 59.28 ±0.88 | 33.13 ±0.22 | 24.50 ±0.25 | 52.63 ±20.51 | 5.69 ±2.76 | 2.46 ±0.24 | 13.43 ±1.06 |
| ReMixMatch | 19.10 ±9.64 | **5.44** ±0.05 | 4.72 ±0.13 | 44.28 ±2.06 | 27.43 ±0.31 | 23.03 ±0.56 | **3.34** ±0.20 | 2.92 ±0.48 | 2.65 ±0.08 | 11.58 ±0.78 |
| EnAET | - | 7.6 ±0.34 | 5.35 ±0.48 | - | - | - | - | 3.21 ±0.21 | 2.92 | - |
| AlphaMatch | 8.65 ±3.38 | **4.97** ±0.29 | - | **38.74** ±0.32 | **25.02** ±0.27 | - | **2.97** ±0.26 | 2.44 ±0.32 | - | **9.64** ±0.75 |
| CoMatch | **6.91** ±1.39 | **4.91** ±0.33 | - | - | - | - | - | - | - | - |
| ReRankMatch | 18.25 ±9.44 | 6.02 ±1.31 | 4.40 ±0.06 | 69.62 ±1.33 | 31.75 ±0.33 | **22.32** ±0.65 | 20.25 ±4.43 | 2.44 ±0.07 | **2.19** ±0.09 | - |
| Meta PL∗ | - | - | **3.89** ±0.07 | - | - | - | - | - | **1.99** ±0.07 | - |
| FixMatch (CTA) | 11.39 ±3.35 | **5.07** ±0.33 | 4.31 ±0.15 | 49.95 ±3.01 | 28.64 ±0.24 | **23.18** ±0.11 | 7.65 ±7.65 | 2.64 ±0.64 | 2.28 ±0.19 | **10.72** ±0.63 |
| CSSL (CTA) | **6.50** ±0.90 | 5.48 ±0.49 | 4.43 ±0.10 | 43.43 ±1.39 | 28.39 ±1.09 | 23.25 ±0.28 | 3.67 ±2.36 | **2.18** ±0.12 | **1.99** ±0.13 | **10.54** ±0.71 |

Table 2: Averaged misclassification rates and expected calibration errors (ECE) using 15 bins for 5 different seeds.

| | CIFAR-10 | | | | SVHN | | | |
| | 40 lab. | | 4000 lab. | | 40 lab. | | 1000 lab. | |
| | Err. | ECE | Err. | ECE | Err. | ECE | Err. | ECE |
|---|---|---|---|---|---|---|---|---|
| FixMatch | 11.39 ±3.35 | 0.087 ±0.051 | **4.31** ±0.15 | 0.030 ±0.002 | **7.65** ±7.65 | **0.040** ±0.044 | 2.28 ±0.19 | 0.010 ±0.002 |
| FixMatch (DA) | **7.73** ±1.92 | 0.048 ±0.012 | 4.64 ±0.10 | 0.027 ±0.001 | **5.21** ±2.85 | **0.031** ±0.020 | **2.04** ±0.38 | 0.010 ±0.001 |
| LSMatch | **8.37** ±1.63 | **0.038** ±0.012 | 5.60 ±1.32 | **0.024** ±0.007 | **3.82** ±1.46 | 0.086 ±0.046 | 2.13 ±0.11 | 0.018 ±0.011 |
| CSSL | **6.50** ±0.90 | **0.032** ±0.005 | **4.43** ±0.10 | **0.023** ±0.001 | **3.67** ±2.36 | **0.022** ±0.029 | **1.99** ±0.13 | **0.007** ±0.001 |

## 4.2 Network Calibration

In a second study, we evaluate FixMatch-based models in terms of network calibration, i.e., the quality of the predicted class probabilities. For this purpose, we calculate the expected calibration error (ECE) as done in [21] using discretized probabilities into 15 bins. The calibration errors provide insight into the bias of the models induced by the different learning methods.

Besides the "raw" hard-labeling, we also consider the distribution alignment (DA)-variant of FixMatch (as described in [41]), as well as an adaptive variant using label smoothing [44], which we dub *LSMatch* (see appendix for an algorithmic description). For label smoothing, we use a uniform distribution policy and calculate the distribution mass parameter $\alpha$ in an adaptive manner as we do for CSSL (cf. Section 3.4). As a result, LSMatch can be regarded as the natural counterpart of our approach for a more cautious learning using less extreme targets. We refer to [29] for a more thorough analysis of the differences between smoothed and credal labels. Since both methods realize an implicit calibration [29, 33], we omit explicit calibration methods that require additional data. Besides, we experiment with an uncertainty-filtering variant of FixMatch following UPS [38], for which we provide results in the supplement.

In accordance with the studies in [29], the results provided in Table 2 show improved calibration compared to classical probabilistic modeling. As these effects are achieved without requiring an additional calibration data split, it provides an appealing method to induce well calibrated and generalizing models. Nevertheless, the margin to the other baselines gets smaller with an increasing number of labeled instances, which is plausible as it implies an increased level of certainty.

## 4.3 Efficiency

In addition to accuracy, a major concern of learning algorithms is run-time efficiency. In this regard, we already noted that thresholding mechanisms may severely delay the incorporation of unlabeled instances in the learning process. For example, for a data set with $2^{26}$ images, FixMatch requires up to $2^{20}$ updates to provide the competitive results reported. This not only excludes potential users without access to computational resources meeting these demands, it also consumes a substantial amount of energy [43].

Table 3: Averaged misclassification rates after 1/8 (CIFAR-10) and 1/32 (SVHN) of the original iterations used for the results in Tab. 1 (**bold** font indicates the single best performing method).

| | CIFAR-10 | | SVHN | |
| --- | --- | --- | --- | --- |
| | 40 lab. | 4000 lab. | 40 lab. | 1000 lab. |
| FixMatch ($\tau = 0.0$) | 18.50 $\pm$2.92 | 6.88 $\pm$0.11 | 13.82 $\pm$13.57 | **2.73** $\pm$0.04 |
| FixMatch ($\tau = 0.8$) | 11.99 $\pm$2.32 | 7.08 $\pm$0.13 | 3.52 $\pm$0.44 | 2.85 $\pm$0.08 |
| FixMatch ($\tau = 0.95$) | 14.73 $\pm$3.29 | 8.26 $\pm$0.09 | 5.85 $\pm$5.10 | 3.03 $\pm$0.07 |
| LSMatch | 11.60 $\pm$2.68 | 7.24 $\pm$0.21 | 7.04 $\pm$3.29 | 2.76 $\pm$0.05 |
| CSSL | **10.04** $\pm$3.32 | **6.78** $\pm$0.94 | **3.50** $\pm$0.49 | 2.84 $\pm$0.06 |

As described before, CSSL allows for incorporating all instances from the very beginning. To measure the implied effects, i.e., a faster convergence, apart from using more effective optimizers, we train the models from the former experiment on CIFAR-10 and SVHN for only an eighth and thirty-second of the original number of epochs, respectively. As we would grant CSSL and LSMatch an unfair advantage compared to confidence thresholding in FixMatch, we experiment with different thresholds $\tau \in \{0, 0.8, 0.95\}$ for FixMatch. The (averaged) learning curves are provided in the supplement.

As shown by the results in Table 3, CSSL achieves the best performance in the label-scarce cases, which confirms the adequacy of incorporating all instances in a cautious manner from the very beginning. Likewise, LSMatch shows competitive performance following the same intuition. In contrast, FixMatch with a high confidence threshold shows slow convergence, which can be mitigated by lowering the thresholding. However, FixMatch with $\tau = 0.0$ provides unsatisfying performance as it drastically increases the risk of confirmation biases; $\tau = 0.8$ seems to be a reasonable trade-off between convergence speed and validity of the model predictions. Nevertheless, when providing higher numbers of labels, the performance gain in incorporating more instances from early on is fairly limited, making the concern of efficiency arguably less important in such cases.

## 5   Conclusion

Existing (probabilistic) methods for self-supervision typically commit to single probability distributions as pseudo-labels, which, as we argued, represents the uncertainty in such labels only insufficiently and comes with the risk of incorporating an undesirable bias. Therefore, we suggest to allow the learner to use credal sets, i.e., sets of (candidate) distributions, as pseudo-labels. In this way, a more faithful representation of the learner's (lack of) knowledge about the underlying ground truth distribution can be achieved, and the risk of biases is reduced. By leveraging the principle of generalized risk minimization, we realize an iterative disambiguation process that implements an implicit trade-off between cautious self-supervision and entropy minimization.

In an exhaustive empirical evaluation in the field of image classification, the enhanced expressiveness and uncertainty-awareness compared to conventional probabilistic self-supervision proved to yield superior generalization performance, especially in the regime of label-scarce semi-supervised learning. Moreover, the experiments have shown an improved network calibration when trained with credal self-supervision, as well as an increased efficiency when considering small compute budgets for training.

Motivated by these promising results, we plan to further elaborate on the idea of credal target modeling and extend it in various directions. For example, as we considered rather simple target sets so far, a thorough investigation of more sophisticated modeling techniques should be conducted. Such techniques may help, for instance, to sift out implausible classes early on, and lead to more precise pseudo-labels without compromising validity. Besides, as already mentioned, our CSSL framework is completely generic and not restricted to applications in image processing. Although we used it to extend FixMatch in this paper, it can extend any other self-training method, too. Elaborating on such extensions is another important aspect of future work.

## Acknowledgments and Disclosure of Funding

This work was supported by the German Research Foundation (DFG) (Grant No. 420493178). Moreover, the authors gratefully acknowledge the funding of this project by computing time provided by the Paderborn Center for Parallel Computing (PC$^2$) and the research group of Prof. Dr. Marco Platzner.

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
