# A   Credal Self-Supervised Learning: Supplementary Material

## A.1   Algorithmic Description of CSSL

Algorithm 1 provides the pseudo-code of the batch-wise loss calculation in CSSL.

---

**Algorithm 1** CSSL with adaptive precisiation $\alpha$

---

**Require:** Batch of labeled instances with degenerate ground truth distributions $\mathcal{B}_l = \{(\boldsymbol{x}_i, p_i)\}_{i=1}^B \in (\mathcal{X} \times \mathcal{Y})^B$, unlabeled batch ratio $\mu$, batch $\mathcal{B}_u = \{\boldsymbol{x}_i\}_{i=1}^{\mu B}$ of unlabeled instances, unlabeled loss weight $\lambda_u$, model $\hat{p} : \mathcal{X} \longrightarrow \mathbb{P}(\mathcal{Y})$, strong and weak augmentation functions $\mathcal{A}_s, \mathcal{A}_w : \mathcal{X} \longrightarrow \mathcal{X}$, class prior $\tilde{p} \in \mathbb{P}(\mathcal{Y})$, averaged model predictions $\bar{p} \in \mathbb{P}(\mathcal{Y})$

1: $\mathcal{L}_l = \frac{1}{B} \sum_{(\boldsymbol{x},p) \in \mathcal{B}_l} H(p, \hat{p}(\mathcal{A}_w(\boldsymbol{x})))$
2: Initialize pseudo-labeled batch $\mathcal{U} = \emptyset$
3: **for** all $\boldsymbol{x} \in \mathcal{B}_u$ **do**
4:    Derive pseudo label $q$ from $\hat{p}(\mathcal{A}_w(\boldsymbol{x}))$, $\tilde{p}$ and $\bar{p}$ acc. to Eq. (6)
5:    Determine reference class $y := \mathrm{argmax}_{y' \in \mathcal{Y}} q(y')$
6:    $\alpha = 1 - q(y) / \sum_{y' \in \mathcal{Y}} q(y')$
7:    Construct target set $Q_y^\alpha$ as in Eq. (2)
8:    $\mathcal{U} = \mathcal{U} \cup \{(\boldsymbol{x}, Q_y^\alpha)\}$
9: **end for**
10: $\mathcal{L}_u = \frac{1}{\mu B} \sum_{(\boldsymbol{x}, Q_y^\alpha) \in \mathcal{U}} \mathcal{L}^*(Q_y^\alpha, \hat{p}(\mathcal{A}_s(\boldsymbol{x})))$
11: **return** $\mathcal{L}_l + \lambda_u \mathcal{L}_u$

---

## A.2   Algorithmic Description of LSMatch

In Algorithm 2, we provide details on the label smoothing variant of FixMatch as investigated in the experiments, which we call *LSMatch*.

---

**Algorithm 2** LSMatch with adaptive distribution mass $\alpha$

---

**Require:** Batch of labeled instances with degenerate ground truth distributions $\mathcal{B}_l = \{(\boldsymbol{x}_i, p_i)\}_{i=1}^B \in (\mathcal{X} \times \mathcal{Y})^B$, unlabeled batch ratio $\mu$, batch $\mathcal{B}_u = \{\boldsymbol{x}_i\}_{i=1}^{\mu B}$ of unlabeled instances, unlabeled loss weight $\lambda_u$, model $\hat{p} : \mathcal{X} \longrightarrow \mathbb{P}(\mathcal{Y})$, strong and weak augmentation functions $\mathcal{A}_s, \mathcal{A}_w : \mathcal{X} \longrightarrow \mathcal{X}$, class prior $\tilde{p} \in \mathbb{P}(\mathcal{Y})$, averaged model predictions $\bar{p} \in \mathbb{P}(\mathcal{Y})$

1: $\mathcal{L}_l = \frac{1}{B} \sum_{(\boldsymbol{x},p) \in \mathcal{B}_l} H(p, \hat{p}(\mathcal{A}_w(\boldsymbol{x})))$
2: Initialize pseudo-labeled batch $\mathcal{U} = \emptyset$
3: **for** all $\boldsymbol{x} \in \mathcal{B}_u$ **do**
4:    Derive pseudo label $q$ from $\hat{p}(\mathcal{A}_w(\boldsymbol{x}))$, $\tilde{p}$ and $\bar{p}$ acc. to Eq. (6)
5:    Determine reference class $y := \mathrm{argmax}_{y' \in \mathcal{Y}} q(y')$
6:    $\alpha = 1 - q(y) / \sum_{y' \in \mathcal{Y}} q(y')$
7:    Construct smoothed target $q'$ with $q'(y) = 1 - \frac{(|\mathcal{Y}|-1) \cdot \alpha}{|\mathcal{Y}|}$ and $q'(y') = \frac{\alpha}{|\mathcal{Y}|}$ for $y' \neq y$
8:    $\mathcal{U} = \mathcal{U} \cup \{(\boldsymbol{x}, q')\}$
9: **end for**
10: $\mathcal{L}_u = \frac{1}{\mu B} \sum_{(\boldsymbol{x}, q') \in \mathcal{U}} H(q', \hat{p}(\mathcal{A}_s(\boldsymbol{x})))$
11: **return** $\mathcal{L}_l + \lambda_u \mathcal{L}_u$

---

## A.3   Evaluation Details

### A.3.1   Experimental Settings

As discussed in the paper, we follow the experimental setup as described in [6]. For a fair comparison, we keep the hyperparameters the same as used within the experiments for FixMatch, which we provide in Table 1. Note that the parameter $\tau$ does not apply to CSSL nor LSMatch, as these approaches do not rely on any confidence thresholding.

Table 1: Hyperparameters as being used within the experiments for CSSL, FixMatch, LSMatch and all other method derivates (if not stated otherwise).

| Symbol | Description | Used value(s) |
|--------|-------------|---------------|
| $\lambda_u$ | Unlabeled loss weight | 1 |
| $\mu$ | Multiplicity of unlab. over lab. insts. | 7 |
| $B$ | Labeled batch size | 64 |
| $\eta$ | Initial learning rate | 0.03 |
| $\beta$ | SGD momentum | 0.9 |
| Nesterov | Indicator for Nesterov SGD variant | True |
| $wd$ | Weight decay | 0.001 (CIFAR-100), 0.0005 (other data sets) |
| $\tau$ | Confidence threshold | 0.95 |
| $K$ | Training steps | $2^{20}$ |

For CTAugment (and later RandAugment as considered in Section A.4.2), we use the same operations and parameter ranges as in reported in [6] for comparability reasons. In case of CTAugment, we keep the bin weight threshold at $0.8$ and use an exponential decay of $0.99$ for the weight updates. In the latter case, we also follow a purely random sampling, that slightly differs from the original formulation in [2]. We refer to [1, 6] for a more comprehensive overview over the methods and their parameters.

### A.3.2  Technical Infrastructure

To put our approach into practice, we re-used the original FixMatch code base[1] provided by the authors for the already available baselines, models, augmentation strategies and the evaluation, and extended it by our implementations. To this end, we leverage TensorFlow[2] as a recent deep learning framework, whereas the image augmentation functions are provided by Pillow[3]. To execute the runs, we used several Nvidia Titan RTX, Nvidia Tesla V100, Nvidia RTX 2080 Ti and Nvidia GTX 1080 Ti accelerators in modern cluster environments. The code related to our work is available at `https://github.com/julilien/CSSL`.

### A.3.3  Efficiency Experiments: Learning Curves

Figure 1 shows the learning curves of the runs considered in the efficiency study in Section 4.3 (averaged over 5 seeds). As can be seen, both CSSL and LSMatch improve the learning efficiency in label-scarce settings by not relying on any form of confidence thresholding. Although LSMatch turns out to converge slightly faster when observing 40 labels from SVHN, the eventual generalization performance of CSSL is clearly superior. For higher amounts of labels, the results are almost indistinguishable.

### A.4  Additional Experiments

### A.4.1  Simple Synthetic Example

To illustrate the disambiguation principle underlying the idea of credal self-supervised learning, we consider a synthetic (semi-supervised) binary classification problem in a one-dimensional feature space. To this end, we sample 25 labeled and 500 unlabeled instances uniformly from the unit interval. As ground-truth, we define the true probability of the positive class by a sigmoidal shaped function.

Provided this data, we train a simple multi-layer perceptron with a single hidden layer consisting of 100 neurons activated by a sigmoid function. As output layer, we use a softmax-activated dense layer with two neurons. To make use of the unlabeled data, we employ self-training with either hard, soft or credal labels for 100 iterations each. In all three cases, we do not apply any form of distribution alignment or confidence thresholding. We use SGD as optimizer with a learning rate of $0.5$ (no

---

[1]The code is publicly available at `https://github.com/google-research/fixmatch` under the Apache-2.0 License.

[2]`https://www.tensorflow.org/`, Apache-2.0 License

[3]`https://python-pillow.org/`, HPND License

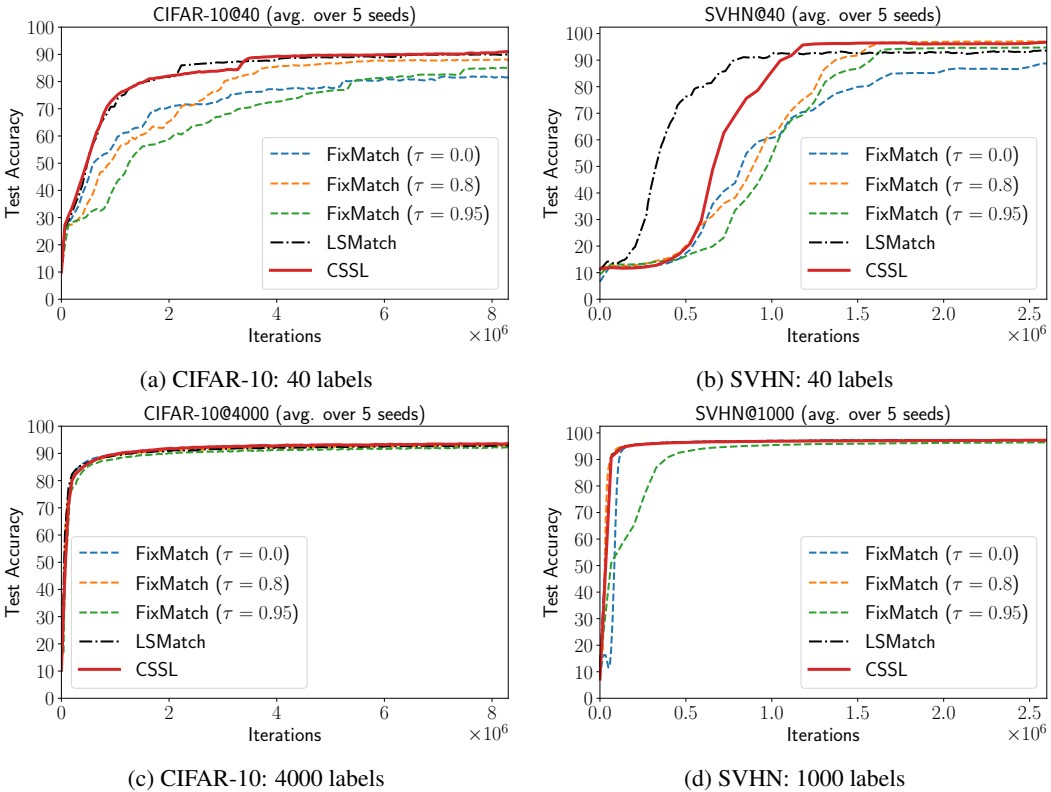

(a) CIFAR-10: 40 labels

(b) SVHN: 40 labels

(c) CIFAR-10: 4000 labels

(d) SVHN: 1000 labels

Figure 1: Averaged learning curves for different dataset configurations.

momentum) for all methods. We repeat each model training (on the same data) 5 times with different seeds.

Fig. 2 shows the labeled and unlabeled data points[4] (red crosses and green circles respectively) and their ground-truth positive class probability (red dashed line), as well as the soft and hard labeling baselines (orange and green lines). The model trained with credal self-supervised learning is indicated by the blue line.

In this setting, self-training of a simple neural network with deterministic labeling leads to a flat (instead of sigmoidal) function most of the time, because the learner tends to go with the majority in the labeled training data. With probabilistic labels, the results become a bit better: the learned functions tend to be increasing but still deviates a lot from the ground-truth sigmoid. Our credal approach yields the best result, being closer to the sigmoid (albeit not matching it perfectly). These results are perfectly in agreement with our intuition and motivation of our approach: Self-labeling examples in an overly "aggressive" (and over-confident) way may lead to self-confirmation and a bias in the learning process.

### A.4.2  Augmentation Ablation Study

In an additional experiment, we perform an ablation study to measure the impact of the augmentation policy on the generalization performance and the network calibration. Here, we compare CSSL with RandAugment and CTAugment as strong augmentation policies. We further distinguish between RandAugment either with or without *cutout* [3], which is a technique to randomly mask out parts of the image. Note that cutout is included in the set of operations we use for CTAugment and was found to be required for strong generalization performance within the FixMatch framework (cf. [6]). Here, we report results on CIFAR-10 for $40$ and $4000$ labels, in both cases trained for $2^{19}$ steps. We use the same hyperparameters as enlisted in Tab. 1 and report the averaged results for 3 seeds.

---

[4]Note that the unlabeled instances have a probability of $0.5$ assigned only for the sake of visualizability, their class distribution also follows the sigmoidal ground-truth.

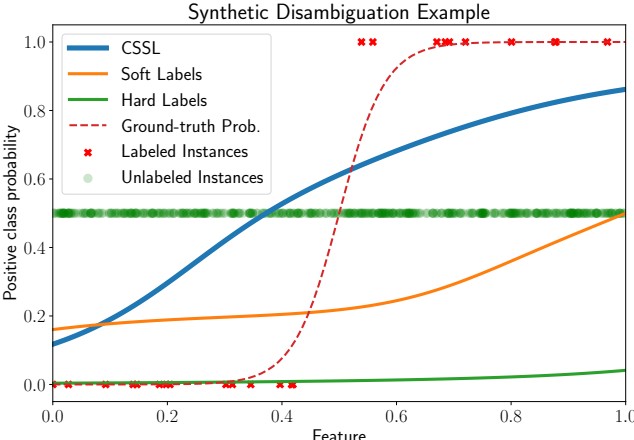

Figure 2: Illustration of CSSL in contrast to conventional probabilistic self-labeling on a simple data-generating process (averaged over 5 seeds).

As can be seen in Table 2, our evaluation confirms the observation in [6] that cutout is a required operation to achieve competitive performance. While RandAugment with cutout as augmentation policy does not show any notable difference compared to CTAugment, employing the former policy without cutout leads to drastically inferior generalization performance and network calibration.

Table 2: Averaged misclassification rates and standard deviations (3 seeds) for various augmentation policies used in CSSL on CIFAR-10 (**bold** font indicates the best performing method and those within a range of two standard deviations from the best method).

| Augmentation | 40 labels | | 4000 labels | |
| --- | --- | --- | --- | --- |
| | Err. | ECE | Err. | ECE |
| CTAugment | **6.92** ±0.34 | **0.035** ±0.005 | **5.11** ±0.64 | **0.028** ±0.001 |
| RandAugment (w/o cutout) | 28.81 ±19.11 | 0.131 ±0.102 | 7.42 ±0.76 | 0.039 ±0.001 |
| RandAugment (w/ cutout) | **6.74** ±0.18 | **0.031** ±0.002 | **5.13** ±0.66 | **0.029** ±0.000 |

## A.5   Barely Supervised Experiments

We also consider the scenario of *barely supervised learning* [6], where a learner is given only a single labeled instance per class. We train all models with $2^{19}$ update iterations on CIFAR-10 and SVHN, using the hyperparameters as described in Section A.3.1. We report the averaged misclassification rates on 5 different folds.

As the uncertainty with such few labels is relatively high, we have experimented with a CSSL version that lower bounds the precisiation degrees $\alpha$, i.e., it defines a minimal size for the target sets. This further increases the degree of cautiousness, but, however, can be seen as an unfair advantage over the other baselines as it adds additional awareness about the highly uncertain nature of the faced problems. In our experiments, we bound the precisiation degree by $\alpha \geq 0.03$, which we determined empirically as a reasonable choice for the two datasets.

Table 3 shows the results. As can be seen, CSSL is able to reduce the error rate compared to conventional hard pseudo-labels in both cases. Moreover, it reduces the intra-dataset variance, which reflects a higher stability of the learning process. Bounding the target set sizes by $\alpha \geq 0.03$ slightly improves the performance on CIFAR-10, but does not show any advantage on SVHN. On the contrary, label smoothing shows an unstable learning behavior in some cases, leading to poor generalization performances for some folds on both data sets. These cases suggest that LSMatch suffers particularly from extremely scarce labeling.

Table 3: Averaged misclassification rates and standard deviations (5 seeds) for the barely supervised experiments with 10 labels (**bold** font indicates the single best performing method).

| Model | CIFAR-10 | SVHN |
|---|---|---|
| FixMatch | 31.74 $\pm$17.77 | 30.57 $\pm$23.82 |
| LSMatch | 42.14 $\pm$25.33 | 44.29 $\pm$27.40 |
| CSSL | 26.90 $\pm$13.57 | **9.69** $\pm$4.63 |
| CSSL ($\alpha \geq 0.03$) | **26.10** $\pm$8.02 | 16.63 $\pm$13.21 |

## A.6 Uncertainty-Based Matching

As the set-valued target modeling in CSSL leads to a form of uncertainty-awareness, one may think of extending classical probabilistic pseudo-labels by additional means measuring the model uncertainty. To this end, UPS [5] augments classical confidence thresholded pseudo-labeling by an additional uncertainty sampling technique (e.g., using MC-Dropout [4]), which is employed in the mechanism to filter out unreliable pseudo-labels. More precisely, it assumes an uncertainty threshold $\kappa$ besides the confidence threshold $\tau$ to calculate the loss on unlabeled instances by

$$\mathcal{L}_u := \frac{1}{\mu B} \sum_{(\boldsymbol{x},q)\in\mathcal{B}_u} \mathbb{I}_{\max \hat{p}(\boldsymbol{x})\geq\tau}\mathbb{I}_{\max u(\hat{p}(\boldsymbol{x}))\leq\kappa}H(q,\hat{p}(\boldsymbol{x})) \ ,$$

where the sum is taken over all unlabeled instances in $\mathcal{B}_u$ with individual pseudo labels $q$ constructed from $\hat{p}(\boldsymbol{x})$, and $u(\cdot)$ is the sampled uncertainty. Note that UPS in its original formulation further uses negative labels, which we omit here for a more the sake of a fair comparison.

We employ the uncertainty-based filtering technique within the framework of FixMatch to compare it with our form of uncertainty-awareness. In the following, we call this variant *UPSMatch*. To induce reasonable thresholds $\tau$ and $\kappa$, we empirically optimize the hyperparameters $\tau$ and $\kappa$ in the spaces $\{0.7, 0.95\}$ and $\{0.1, 0.25, 0.5\}$ respectively on a separate validation set. For MC-Dropout, we use 8 sampling iterations (as opposed to 10 in the original approach) due to computational resource concerns, whereby the drop rate for Dropout is set to 0.3 as also used in [5]. Moreover, we use hard pseudo-labels as in UPS and FixMatch.

Tab. 4 shows the results for UPSMatch and baselines for 5 different seeds after $2^{20}$ training steps. On CIFAR-10, UPSMatch improves over FixMatch for 40 labels in terms of generalization performance and network calibration. This is reasonable as UPSMatch implements a more cautious learning behavior by augmenting the pseudo-label selection criterion and reduces the risk of confirmation biases. Similarly, UPSMatch outperforms FixMatch for 40 labels on SVHN, whereas it provides both worse generalization performance and better calibration in the other cases. Nevertheless, CSSL turns out to be superior compared to both selective methodologies. However, let us emphasize that a more thorough investigation of the interaction between uncertainty-based sampling and consistency regularization as employed in FixMatch needs to be performed, as well as a more sophisticated hyperparameter optimization.

Table 4: Averaged misclassification rates and expected calibration errors (ECE, 15 bins) for 5 seeds (**bold** font indicates the best performing method and those within a range of two standard deviations from the best method).

| | CIFAR-10 | | | | SVHN | | | |
|---|---|---|---|---|---|---|---|---|
| | 40 lab. | | 4000 lab. | | 40 lab. | | 1000 lab. | |
| | Err. | ECE | Err. | ECE | Err. | ECE | Err. | ECE |
| FixMatch | 11.39 $\pm$3.35 | 0.087 $\pm$0.051 | **4.31** $\pm$0.15 | 0.030 $\pm$0.002 | **7.65** $\pm$7.65 | **0.040** $\pm$0.044 | 2.28 $\pm$0.19 | 0.010 $\pm$0.002 |
| CSSL | **6.50** $\pm$0.90 | **0.032** $\pm$0.005 | **4.43** $\pm$0.10 | **0.023** $\pm$0.001 | **3.67** $\pm$2.36 | **0.022** $\pm$0.029 | **1.99** $\pm$0.13 | **0.007** $\pm$0.001 |
| UPSMatch | 10.48 $\pm$2.11 | 0.058 $\pm$0.010 | 4.92 $\pm$0.39 | 0.027 $\pm$0.003 | **3.81** $\pm$1.99 | **0.025** $\pm$0.027 | 2.71 $\pm$0.47 | **0.008** $\pm$0.001 |