# OpenReview forum: "Credal Self-Supervised Learning"
_NeurIPS.cc/2021/Conference — NeurIPS 2021 Poster_

### Official Review · Reviewer_jEBP · 2021-07-15

**Rating:** 6
**Confidence:** 3

**Summary:**

This work studied the credal sets, i.e., sets of probability distributions, in self-supervised learning. The author claimed that the increased expressiveness can improve performance and reduce biases. The proposed method is evaluated empirically on the image classification task.

**Limitations And Societal Impact:**

The author discussed some limitations of this work through experiments in Section 4.

**Main Review:**

## Originality

The use of such credal probability and so-called "possibility distributions" seems rare in machine learning. The proposed method for learning from credal labels is based on the minimal value of the KL-divergence between probability in the set and the prediction, which originated from existing work (optimistic superset loss, infimum loss). The author then used credal labels in self-supervised learning with FixMatch.

To the best of my knowledge, although its components are from existing work (credal label, infimum loss, FixMatch), the application of learning from credal labels in self-supervised learning is new.

---

## Soundness

My major concern is about the usefulness of the concept of credal sets in this setting. Indeed, hard labels may cause problems in neural network training. The author suggested that soft labels, probability distributions, are still not enough. However, the evidence is not convincing enough.

In traditional statistical learning, we want to learn a decision function that outputs labels, where both training data and prediction are deterministic ("hard label"). Since it's NP-hard to optimize it, we can resort to probabilistic models that output probabilities of labels, where the training data is still deterministic. It is possible because the expectation over the empirical distribution converges to the expectation over the true distribution as the size of the sample increases. There are a lot of attempts to let the model provide more uncertainty measures using, for example, Bayesian methods. The supervision may be still deterministic.

The direction of this paper is different from the ones above. Here, the uncertainty of the supervision is increased, from labels to probabilities of labels, to sets of probabilities of labels. Meanwhile, if I understand correctly, the output is still a single-point probability. My concern is, is the same effect can be achieved by, for example, regularizations of the weight or the output? Another straightforward approach would be using distributions over distributions of categorical labels, such as Dirichlet distribution, as the format of supervision. What's the advantage of the proposed method using credal labels?

---

## Technical quality

Another issue is that this method is verified solely empirically. Theoretical characterization is limited or from existing work.

**Time Spent Reviewing:**

4

---

> ### Author Response · Authors · 2021-08-09
> **Reviewer jEBP response**
>
> Thanks a lot for your constructive feedback and acknowledging the originality of our work.
>
> As for the usefulness of the credal approach, we think there is a misunderstanding: We don’t make certain data uncertain, but just the other way around. Like other methods for self-training, we start from unlabeled and hence completely uncertain data, and then successively reduce the uncertainty (by shrinking the credal set). The main difference is that our uncertainty reduction is less aggressive. If the learner assigns a deterministic label to an unlabeled data point, it (implicitly) pretends full certainty about the correctness of this label, which is clearly unwarranted. Even if the learner assigns a probability distribution, it is unlikely to match the ground truth distribution, thereby causing a bias. By labeling data points with a SET of distributions, there is a good chance that the true distribution is contained. Thus, compared to a single distribution, a credal set provides weaker but at the same time more correct information.
>
> You are completely right that second-order distributions such as Dirichlet could in principle be used to model beliefs akin to ours. However, compared to sets of distributions, distributions of distributions introduce yet another level of complexity. Therefore, we would rather see them as a natural next step. Moreover, note that efficient learning from second-order distributions is non-trivial. In our case, we can rely on methods for superset learning, which provides a convenient yet efficient learning framework.
>
> Concerning regularization: Yes, commonly used weight / output regularization methods could be employed to achieve similar cautiousness effects. However, our methodology describes a data modeling technique that provides a more faithful representation of the learner’s beliefs about the underlying ground truth, and should be distinguished from data modeling-agnostic regularization.

---

> > ### Comment · Reviewer_jEBP · 2021-08-10
> > **Thank you for your explanation**
> >
> > Thank you for your explanation of the credal approach, which improved my understanding.
> >
> > I'll look into it and reread the paper and other reviews soon to decide if I should increase my rating. Thank you.

---

### Official Review · Reviewer_AL5C · 2021-07-15

**Rating:** 6
**Confidence:** 3

**Summary:**

This paper proposes to modify the pseudo-labeling procedure in semi-supervised learning. Student model predictions are penalized by distance to a credal set of teacher predictions. The size of credal sets is determined heuristically. Experiments demonstrate improvement over baselines mainly in scarce-label scenarios.

**Limitations And Societal Impact:**

Yes.

**Main Review:**

The central idea is clear, and the experiments demonstrate improvements over the SOTA method Fixmatch. However, there are a few weaknesses / questions:

1. The techniques seem like empirical hacks and lacks theoretical support. Can the authors demonstrate the advantage of the proposed method on a simple data generative model?
2. No ablations on the choice of $\alpha_i$. Equation 6 seems especially ad hoc and requires justification.
3. An important missing baseline is 'Sinkhorn Label Allocation: Semi-Supervised Classification via Annealed Self-Training' by Tai et al.
4. It is not clear to me why distributional matching doesn't work, whereas credal self-training does.

**Time Spent Reviewing:**

2

---

> ### Author Response · Authors · 2021-08-09
> **Reviewer AL5C response**
>
> Thank you very much for your constructive feedback, which we really appreciate. In the following, we address your points one by one:
>
> 1.& 2.: We see your point and agree that Eq. (6) might be considered ad hoc to some extent. That said, one should differentiate between the general methodology and its concrete instantiation. As for the former, we would argue that our methodology of “credal self-supervision” is not only intuitively plausible and well-motivated but also theoretically supported by formal properties of superset learning (e.g. [1]). Like any other methodology, a concrete instantiation requires certain design decisions, such as the weighing scheme -- comparable, for example, to the choice of a kernel in SVMs or a prior in Bayesian inference. It’s very difficult to justify such choices theoretically, let alone proving their optimality. Note, however, that we nevertheless provide a justification of the weighting (6), which is inspired by the distribution alignment in the semi-supervised approach ReMixMatch [2].
>
> An illustrative example with a simple data-generating process is a good idea, thanks for the suggestion. We devised such an example, where instances are real numbers in the unit interval (one-dimensional instance space), and the true probability of the positive class is a sigmoidal shaped function. In the experiment, we generate 500 unlabeled and 25 labeled examples. In this setting, self-training of a simple neural network with deterministic labeling leads to a flat (instead of sigmoidal) function most of the time, because the learner tends to go with the majority in the labeled training data. With probabilistic labels, the results become a bit better: the learned functions tend to be increasing but still deviates a lot from the ground-truth sigmoid. Our credal approach yields the best result, being close to the sigmoid (albeit not matching it perfectly). These results are perfectly in agreement with our intuition and motivation of our approach: Self-labeling examples in an overly “aggressive” (and over-confident) way may lead to self-confirmation and a bias in the learning process. We cannot show the results graphically here, but we’ll include them in the supplement of the paper.
>
> 3.: Thank you for the reference, which is indeed related to our work. However, let us note that this paper is very recent. It has been published at ICLR this year, contemporaneous to the NeurIPS 2021 submission deadline (at least according to the reviewing guidelines of NeurIPS 2020). Even the arXiv version was available only a bit earlier. Nevertheless, we plan to address this reference in a camera-ready version of our work.
>
> 4.: We assume by distributional matching you mean distribution alignment as introduced for ReMixMatch [2]. Well, we wouldn’t say that it doesn’t work. In Table 2, one can see clear improvements in terms of generalization performance and calibration using DA within FixMatch. Thus, it works better as the unaligned target modeling. Regarding the comparison of our method to DA, the results are quite in agreement with previous comparisons in [3], where advantages of credal target modeling over single distributions have shown in a related setting. As it is explained in the paper, the main reason for the improvement is the avoidance of a bias caused by the choice of a single distribution that deviates from the ground truth. Provided with a credal, the learner has the freedom to select the most plausible distribution itself, i.e., the distribution is chosen in a data-driven way instead of being pre-defined.
>
> [1] Cabannes et al. Structured Prediction with Partial Labelling through the Infimum Loss. ICML 2020.
>
> [2] Berthelot et al. ReMixMatch: Semi-Supervised Learning with Distribution Alignment and Augmentation Anchoring. CoRR abs/1911.09785, 2019.
>
> [3] Lienen and Hüllermeier. From Label Smoothing to Label Relaxation. AAAI 2021.

---

> > ### Comment · Reviewer_AL5C · 2021-09-02
> > **Thank you for your response**
> >
> > Thank you for your clarifications. I agree that this paper made empirically progress so I will raise my score, but I'm not convinced that it is the theory that yields the performance gains. I think better comparisons with related works and ablations would enhance the paper.

---

### Official Review · Reviewer_TVWJ · 2021-07-16

**Rating:** 7
**Confidence:** 3

**Summary:**

For the task of semi-supervised learning, this work proposes the use of credal sets to model uncertainty in pseudo-labels. By proposing simple scheme for generating and learning from these credal sets, the authors show how the FixMatch baseline can be extended and improved in settings where the number of labeled examples is small. Furthermore, they show that this approach can improve the efficiency of this semi-supervised learning approach and reduce calibration errors.

**Ethical Concerns:**

No ethical concerns.

**Limitations And Societal Impact:**

The authors do not discuss limitation or societal impacts at all in their work, this is unfortunate as it represents and area where the authors could point to the importance of their work in various real-world settings.

**Main Review:**

The authors proposed used of credal sets to model uncertainty in the pseudo-labeling pipeline of semi-supervised learning is original, well-motivated, and demonstrated to be effective. The authors clearly motivate their approach and through description and accompanying pseudo code present the work in a manner than can be replicated and re-implemented. As both semi-supervised learning and uncertainty quantification are important and timely areas of interest within the machine learning community, this work has significance based on its ability to successfully combine important approaches in both areas.

The paper seems to lack ablative experiments around various aspects of CSSL, such as exponential moving averaging, which would help clarify which portions of the proposed approach contributed to the improved performance the most significantly.

The paper is well-written, addresses an important shortcoming of existing ssl approaches (efficiency and generalization) a and presents a way to improve upon competitive baselines.

**Time Spent Reviewing:**

6

---

> ### Author Response · Authors · 2021-08-09
> **Reviewer TVWJ response**
>
> Thank you very much for the positive feedback. You are right, further ablation studies could be done, and would probably give more insights into the contributions of the different components of CSSL. For the assessment of our method, we employed credal self-supervised learning in the fundamental framework of FixMatch, which features well-explored state-of-the-art design choices for semi-supervised learning models. This is why, especially in light of the computational complexity of the experiments, our empirical evaluation focuses more on the differences between conventional probabilistic and credal target modeling in various scenarios.
>
> Regarding the limitations of our method proposal, we refer to the description of the empirical results, where we point out that our method mainly improves in label-scarce settings.
>
> As for societal impact, you are right, our method certainly has a high potential for real-world settings. Yet, as it is still in an early phase of development, we didn’t want to overclaim. Anyway, we plan to add something on this point in the camera-ready version.

---

### Official Review · Reviewer_qetJ · 2021-07-18

**Rating:** 7
**Confidence:** 4

**Summary:**

Unlabelled samples are labelled with credal sets (set of probability distributions), better capturing the 'unknowledge' about the label.

The methodology integrates seamlessly several state-of-the-art ideas to achieve top performance in the experimental evaluation.



**Limitations And Societal Impact:**

yes.

**Main Review:**

The paper is well-written and technically sound.
The novelty of the paper is enough.
Although the improvements in accuracy are not impressive, the contribution can be of interest to a wide range of NeurIPS participants.

The ablation study could include an analysis of the importance of the weighting in Eq (6) to the final results. What would be the results with the straightforward alphas?

**Time Spent Reviewing:**

2

---

> ### Author Response · Authors · 2021-08-09
> **Reviewer qetJ response**
>
> Thank you very much for your appreciation and kind words. We indeed experimented with a non-weighted $\alpha$ imprecisiation in the first place, but observed an increased risk of confirmation biases. Without the distribution alignment change, models tend to reduce guesses of $\alpha$ for unlabeled instances too quickly.

---

### Decision · Program_Chairs · 2021-09-27

**Decision:**

Accept (Poster)

**Comment:**

After the discussion, the reviewers agree that the paper proposes an interesting approach of using credal sets as a generalization of using probability estimates in self-supervision approaches. The resulting loss function is relatively simple and the authors describe one practical way to implement it.

The paper is still borderline because the experimental results, even though they seem slightly better than the initial FixMatch approach, are not significantly better than other baselines, and additional ablations could be performed regarding some design choices, such as the effect of the weighting in Eq 6 or the exponential moving average of model weights. These ablations would be good to have, but since these come from previous work on the topic and the baselines are included in the main table, I do not feel that these are decisive arguments for rejection, even though they would be good to have.

Overall, the approach proposed in the paper provides a cleaner way of using pseudo-labeling than the confidence matching, which is illustrated for instance in section on efficiency (learning curves might be better than a fixed number of epochs here), where we see the advantage of not requiring the confidence threshold as additional hyperparameter. In that respect, the approach can be used in practice even if it does not significantly improves performance compared to the optimally tuned competitors.